# DistillMatch: Leveraging Knowledge Distillation from Vision Foundation Model for Multimodal Image Matching

## Abstract

Multimodal image matching seeks pixel-level correspondences between images of different modalities, crucial for cross-modal perception, fusion and analysis. However, the significant appearance differences between modalities make this task challenging. Due to the scarcity of high-quality annotated datasets, existing deep learning methods that extract modality-common features for matching perform poorly and lack adaptability to diverse scenarios. Vision Foundation Model (VFM), trained on large-scale data, yields generalizable and robust feature representations adapted to data and tasks of various modalities, including multimodal matching. Thus, we propose DistillMatch, a multimodal image matching method using knowledge distillation from VFM. DistillMatch employs knowledge distillation to build a lightweight student model that extracts high-level semantic features from VFM to assist matching across modalities. To retain modality-specific information, it extracts and injects modality category information into the other modality's features, which enhances the model's understanding of cross-modal correlations. Furthermore, we design V2I-GAN to boost the model's generalization by translating visible to pseudo-infrared images for data augmentation. Experiments show that DistillMatch outperforms existing algorithms on public datasets.

## 1 Introduction

Multimodal images, like visible and infrared images from different sensors, can provide richer scene information Jiang et al. (2021); Zhou et al. (2022). They are crucial for advanced visual tasks including medical image analysis Li et al. (2025), remote sensing image processing Xiao et al. (2024); Li et al. (2019), and autonomous driving Zhou et al. (2021). However, the variations in imaging positions lead to geometric normalization issues in multimodal images, such as scale, rotation, and viewpoint changes, making precise analysis difficult for computers. Multimodal image matching enhances the accuracy and robustness of visual tasks by establishing correspondences across modalities, thereby promoting the development of multimodal perception technologies and expanding its application.

The imaging principles of Multimodal images are distinct, leading to significant discrepancies in texture, contrast, and intensity Tang et al. (2022); Li et al. (2013). These modal differences reduce the feature extraction accuracy and limit the effectiveness of traditional matching methods. Current deep-learning methods focus on extracting modality-common features for matching, discarding modality-specific information and limiting feature representation Hou et al. (2024); Shi et al. (2023); Deng et al. (2023); Liu et al. (2024); Deng & Ma (2023). Besides, due to the scarcity of large-scale, high-quality annotated datasets, models are mostly trained on single-modality and small-scale unannotated multimodal datasets, resulting in poor generalization and adaptability to diverse scenarios, which restrict the practical application of multimodal image matching.

To tackle these issues, we propose DistillMatch for multimodal image matching via knowledge distillation from VFM in Figure 1 (a). VFM like DINOv2 Oquab et al. (2023), trained on extensive data, can extract high-level, modality-independent semantic features, which are resistant to modal differences and noise. Basic feature extractors yield texture features with local geometric information for matching, which are not robust to modal differences. Thus, we use features from VFM

Figure 1: Overview of DistillMatch, CEFG and STFA. (a) Framework of DistillMatch includes KD-VFM and matching module. (b) Structure of CEFG module. (c) Structure of STFA module.

to guide extractor to focus on semantically similar regions. DistillMatch transfers VFM's semantic knowledge into a lightweight student model via online knowledge distillation, which inherits semantic understanding and adapts to matching tasks. To retain modality-specific information, we design a Category-Enhanced Feature Guidance Module (CEFG) that injects modality category representation from one modality into another's features, enhancing texture features' understanding of cross-modal correlations. Then, STFA aggregates semantic and enhanced texture features to integrate their advantages. For matching, a coarse-to-fine matching module is used to establish subpixel-level correspondences. To address data scarcity, we propose V2I-GAN for visible-to-infrared image translation for data augmentation. Extensive experiments on public datasets show DistillMatch outperforms state-of-the-art algorithms. Anonymized source code can be found in the Reproducibility Statement. The paper has the following contributions:

- We design a lightweight student model that uses online knowledge distillation to learn high-level semantic understanding from VFM, overcoming modal differences.

- We design a Category-Enhanced Feature Guidance Module. It injects modality category representation to enhance understanding of cross-modal correlations.

- We propose V2I-GAN for visible-to-infrared image translation, overcoming the limited training data issue.

## 2 RELATED WORKS

### 2.1 DATA AUGMENTATION BASED MATCHING METHODS

To address the scarcity of annotated data in multimodal image matching, researchers used the methods of data augmentation Deng et al. (2024); Zhang et al. (2025a). They generate high-quality synthetic or pseudo-multimodal datasets for mixed training to boost performance Zhu et al. (2017); Han et al. (2024). He et al. proposed a general large-scale pre-training framework for data augmentation He et al. (2025), that integrates cross-modal signals from various data sources, enabling model to recognize and match fundamental image structures. Jiang et al. introduced MINIMA, a unified image matching framework Ren et al. (2025). They designed a data engine to expand single-modal RGB images into multimodal data and built a new MD-syn dataset. MD-syn can directly train any advanced matching pipeline, significantly improving their performance in multimodal matching. Liu et al. constructed a real infrared-visible image dataset MTV Liu et al. (2022), using UAV-captured images, 3D reconstruction technology, and semi-supervised generation methods, and retrained LoFTR Sun et al. (2021) for multimodal matching.

### 2.2 PRE-TRAINED AND FINE-TUNED MATCHING METHODS

To overcome modal differences and extract cross-modal high-level features, researchers pre-train feature extractors on large-scale data and fine-tune them for multimodal matching Zhou et al. (2022); Yagmur et al. (2024). Pre-training doesn't require datasets with matching annotations, and can use data from other domains, reducing data collection and annotation costs. Tuzcuoğlu et al. proposed XoFTR Tuzcuoğlu et al. (2024), which uses masked image modeling for pre-training and fine-tunes with pseudo-infrared images. Zhang et al. introduced SemaGlue Zhang et al. (2025b), which combines semantic information from pre-trained segmentation model and image geometric features, enhancing semantic understanding in matching. Zhang et al. proposed SDME Zhang & Ma (2024),

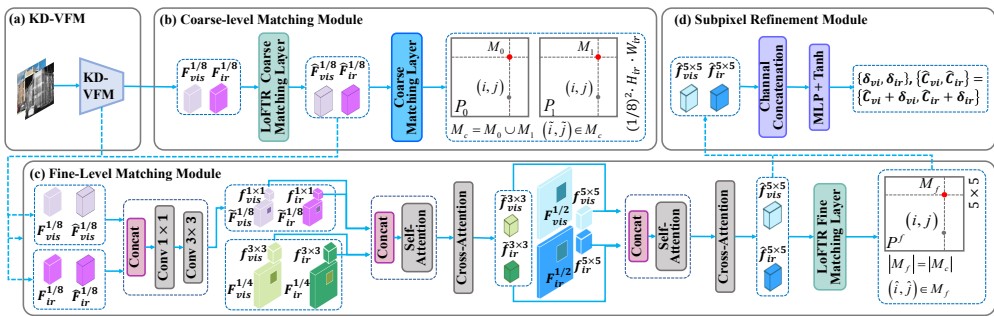

Figure 2: Overview of matching module. (a) is KD-VFM. (b) is coarse-level matching module, which predicts coarse-level matches at the 1/8 scale. (c) is fine-level matching module, which uses 1/2 and 1/4 scale features based on the coarse-level matches to predict fine-level matches. (d) is subpixel refinement module, which refines fine matches at the subpixel level.

which performs initial registration via sparse feature matching prediction and refines results through dense direct alignment. It can fine-tune model pre-trained on single-modal datasets using small multimodal datasets. Sun et al. proposed DenseAffine for extracting affine correspondences Sun et al. (2025), introducing a geometry-constrained loss function combined with dense matches to supervise networks in learning geometric information. DenseAffine uses ResNet50 He et al. (2016) encoder pre-trained on ImageNet-1K Deng et al. (2009), fine-tuning only the Refiner module's weights. Liu et al. proposed LiftFeat Liu et al. (2025), a lightweight network that uses pseudo surface normal labels from pre-trained monocular depth estimation model to extract 3D geometric feature. It enhances 2D feature description discrimination by fusing 3D with 2D descriptors.

### 2.3 VFM Based Matching Methods

VFM, trained on large-scale image datasets, excels in representation and semantic understanding Xue et al. (2023); Edstedt et al. (2024); Zhang & Zhao (2024); Xue et al. (2025). Many researchers use VFM to capture cross-modal semantic features for matching, overcoming modal differences and reducing reliance on large-scale annotated data. Cadar et al. proposed SCFeat Cadar et al. (2024), enhancing local feature matching with semantic features from model like DINOv2. It optimizes descriptors by fusing texture and semantic features through a semantic reasoning module. Wu et al. introduced SAMFeat Wu et al. (2023), which uses the Segment Anything Model (SAM) Kirillov et al. (2023) as a teacher model. Through knowledge distillation, contrastive learning, and edge attention guidance, SAMFeat extracts semantic information from SAM to optimize local feature descriptors. Lu et al. proposed JamMa Lu & Du (2025), an ultra-lightweight feature matching method based on joint Mamba. Using the linear Mamba Gu & Dao (2023) model and JEGO scan-merge strategy, it achieves efficient image matching.

## 3 Methodology

DistillMatch has four modules: KD-VFM, CEFG, STFA module, and matching modules from coarse to fine. We also propose an image translation method V2I-GAN for data augmentation.

### 3.1 Feature Extraction Module based on Knowledge Distillation of VFM

To leverage the high-level semantic cues from VFM, we design the KD-VFM module, which can aggregate high-level semantic information into basic feature extractors. The structure of KD-VFM is shown in Figure 1 (a).

**Feature Extraction:** Given two multimodal images from the same scene, e.g., visible and infrared image $I_{vis/ir}$, they are input to KD-VFM. KD-VFM has three different branches. The first branch is a multibranch and multiscale ResNet, which processes $I_{vis/ir}$ and generates basic texture features $F_{Res}^{1/2} \in \mathbb{R}^{B \times C_1 \times \frac{H}{2} \times \frac{W}{2}}$, $F_{Res}^{1/4} \in \mathbb{R}^{B \times C_2 \times \frac{H}{4} \times \frac{W}{4}}$ and $F_{Res}^{1/8} \in \mathbb{R}^{B \times C_3 \times \frac{H}{8} \times \frac{W}{8}}$, where $H$ and $W$ are image's height and width, and $C_1 = 128, C_2 = 196, C_3 = 256$. The second branch uses a ViT-S/14 variant of the DINOv2 model augmented with register tokens. It generates high-level semantic

features $F_{DINO}$. Prior to feeding images into this branch, they are downsampled to $7/8$ of original resolution. The output $F_{DINO} \in \mathbb{R}^{B \times C_4 \times \frac{H}{14} \times \frac{W}{14}}$ are interpolated to the $1/8$ of original resolution using bilinear interpolation to obtain $F_{DINO} \in \mathbb{R}^{B \times C_4 \times \frac{H}{8} \times \frac{W}{8}}$, where $C_4 = 384$.

**Distillation of VFM:** DINOv2 is a Transformer-based pretrained VFM trained on large-scale datasets with strong generalization and can capture rich and robust semantic information from images. However, its complex architecture leads to high computation and slow inference, limiting deployment in resource-constrained scenarios. To solve this and avoid loading DINOv2's pretrained weights, we propose a lightweight vision transformer Dosovitskiy et al. (2021) as student model in the third branch, trained to distill knowledge from the teacher model's output $F_{tea} = F_{DINO}$. In multimodal image matching, different modalities have different feature distributions and noise characteristics. Though $F_{DINO}$ has broad generalizability, it may not fully adapt to domain-specific matching scenarios, potentially underutilizing task-relevant information. Thus, we propose an online feature distillation framework. The student model is fine-tuned on task-specific datasets and losses, enabling it to learn matching-oriented features in training, enhancing algorithmic stability.

In student model, the input image is divided into fixed-size patches and embedded into a high-dimensional embedding space with 2D sinusoidal-cosine positional encoding to generate initial feature $F_P \in \mathbb{R}^{B \times P \times C_4}$, where $P = 1600$. The encoder has multiple transformer blocks, each with a multi-head self-attention layer and a feed-forward network, while the decoder has a similar structure. The model ultimately outputs the refined feature $F_{stu} \in \mathbb{R}^{B \times C_4 \times \frac{H}{8} \times \frac{W}{8}}$.

To effectively distill high-quality features from DINOv2 to student model, we design a comprehensive feature alignment loss that integrates three methods. The mean squared error (MSE) loss quantifies the discrepancy between $F_{stu}$ and $F_{tea}$ using MSE:

$$L_{MSE} = \frac{1}{N} \sum_{i=1}^{N} \left\| \frac{F_{tea}}{\|F_{tea}\|_2} - \frac{F_{stu}}{\|F_{stu}\|_2} \right\|_2^2, \tag{1}$$

where $N = BHW/64$ is the dimensionality of the flattened feature vectors. $L_{MSE}$ enforces numerical proximity between $F_{stu}$ and $F_{tea}$ at the pixel level.

Gram matrix loss quantifies feature similarity by comparing the Gram matrices of $F_{stu}$ and $F_{tea}$:

$$L_{Gram} = \frac{1}{N} \sum_{i=1}^{N} \|G(F_{tea}) - G(F_{stu})\|_2^2, \tag{2}$$

where $G(F) = \frac{FF^T}{HW}$. $N$ is the number of elements in Gram matrix. $L_{Gram}$ enforces spatial-relationship preservation between $F_{stu}$ and $F_{tea}$.

The Kullback-Leibler (KL) divergence loss quantifies discrepancy in the probabilistic distribution between $F_{stu}$ and $F_{tea}$:

$$L_{KL} = D_{KL}(F_{tea} \| F_{stu}), \tag{3}$$

where $D_{KL}(\cdot)$ is the KL divergence operator. $L_{KL}$ enforces probabilistic distribution alignment between $F_{stu}$ and $F_{tea}$.

The complete knowledge distillation loss is formulated as:

$$L_{KD} = \alpha \cdot L_{MSE} + \beta \cdot L_{Gram} + \gamma \cdot L_{KL}, \tag{4}$$

where $\alpha$, $\beta$ and $\gamma$ are the weights.

## 3.2 Category-Enhanced Feature Guidance Module

Modal differences cause the texture features extracted by KD-VFM exhibiting significant divergence, making it hard to establish correspondence of them across same regions in different modalities. To mitigate modal differences and enhance the understanding of cross-modal correlations, we propose the Category-Enhanced Feature Guidance Module (CEFG). As shown in Figure 1 (b), CEFG uses an encoder composed of restormer and transformer layers. The restormer processes input $F_{Res}^{1/2}$, and produces shallow features $F_{vis/ir}^s \in \mathbb{R}^{B \times N \times C_3}$ ($N = 1600$), which contain low-level image details. We initialize a learnable category feature $f_{vis/ir}^{clc} \in \mathbb{R}^{B \times 1 \times C_3}$ and concatenate it with $F_{vis/ir}^s$. The combined features are then processed through two transformer layers

and split into deep-level features $\widehat{F}^d_{vis/ir} \in \mathbb{R}^{B \times N \times C_3}$ and modality category representation heads $\widehat{f}^{clc}_{vis/ir} \in \mathbb{R}^{B \times 1 \times C_3}$. $\widehat{f}^{clc}_{vis/ir}$ is used to characterizes the image's modality category. To ensure that $\widehat{f}^{clc}_{vis/ir}$ precisely represents the modality-aware information, we use MLP and optimize it with cross-entropy loss $L_{ce}$:

$$L_{ce} = CE\left(P_{vis}, [0,1]\right) + CE\left(P_{ir}, [1,0]\right) \tag{5}$$

where $CE(\cdot)$ is the cross-entropy function and $P_{vis/ir}$ is the MLP prediction. $L_{ce}$ enforces the MLP's output to accurately predict modality category labels.

As modal difference persists between $\widehat{F}^d_{vis}$ and $\widehat{F}^d_{ir}$, they cannot be directly matched. Conventional methods extract common cross-modal features from them for matching, but they discard modality-specific or non-shared information, compromising the feature representational capacity. To solve this, we directly inject $\widehat{f}^{clc}_{ir/vis}$ as global feature information into $\widehat{F}^d_{vis/ir}$ through element-wise summation, and then input them separately into two Transformer blocks with non-shared parameters (TransferA and TransferB) to obtain the category-enhanced feature $\widetilde{F}_{vis/ir}$: $\widetilde{F}_{vis} = TransferA(\widehat{F}^d_{vis} + \widehat{f}^{clc}_{ir}), \widetilde{F}_{ir} = TransferB(\widehat{F}^d_{ir} + \widehat{f}^{clc}_{vis})$.

To guide texture features through category-enhanced features, we directly fuse $\widetilde{F}_{vis/ir}$ with $F^{1/8}_{Res}$ via element-wise addition, and input it into convolutional layers to obtain enhanced texture features $\widetilde{F}^{1/8}_{Res}$. This operations not only enhance the model's comprehension of cross-modal correlations but also preserve non-shared information.

### 3.3 SEMANTIC AND TEXTURE FEATURE AGGREGATION MODULE

Texture feature $F_T = \widetilde{F}^{1/8}_{Res}$ excels at capturing local geometric information but lacks semantic comprehension. Semantic feature $F_S = F_{stu}$ demonstrates strong scene-level semantic understanding yet suffers from insufficient resolution for fine-level matching. To aggregate the strengths of both features and enhance representational capacity and matching precision, we design the Semantic and Texture Feature Aggregation Module (STFA), which contains Channel Attention Aggregation (CAA) module and Spatial Attention Aggregation (SAA) module.

As shown in Figure 1 (c), the CAA module first aligns the channel and spatial dimensions of $F_S$ with $F_T$ by bilinear interpolation and channel compression. The aligned features are then reshaped and input to MLP and layer-normalization, yielding $F^{LN}_{S/T} = LN\left(MLP(F_{S/T})\right) \in \mathbb{R}^{B \times N \times C}$, where $LN(\cdot)$ is layer-normalization. Finally, $F^{LN}_S$ is used as the query, and $F^{LN}_T$ is used as the key and value to perform cross-attention aggregation along the channel dimension and obtain $F^{CAA}_T$. CAA achieves soft channel-dimension alignment, enabling semantic features to adaptively focus on channels relevant to texture features, thereby enhancing feature consistency.

SAA has a similar structure to CAA. First, $F^{CAA}_T$ and $F_T$ are fed into convolutional projection layers to generate: $Q = Proj_q(F_T), K = Proj_k(F^{CAA}_T), V = Proj_v(F^{CAA}_T) \in \mathbb{R}^{B \times C \times N}$. Then perform spatial attention aggregation along the spatial dimension and obtain $F^{SAA}_T$. Finally, perform residual connection between $F^{SAA}_T$ and the original feature to obtain $F^{STFA}_T = F_T + reshape(F^{SAA}_T) \in \mathbb{R}^{B \times C \times \frac{H}{8} \times \frac{W}{8}}$. SAA enables texture features to acquire spatially relevant information from semantic features, achieving feature fusion.

### 3.4 MATCHING MODULE FROM COARSE TO FINE

**Coarse-level Matching Module (CMM):** CMM uses feature $F^{1/8}_{vis}$ and $F^{1/8}_{ir}$ from STFA to predict matches at the 1/8 scale. As shown in Figure 2 (b), it first applies linear self-attention and cross-attention in LoFTR to interact $F^{1/8}_{vis}$ and $F^{1/8}_{ir}$, outputting $\hat{F}^{1/8}_{vis}$ and $\hat{F}^{1/8}_{ir}$. The similarity matrix $S$ is computed as: $S(i,j) = \frac{1}{\gamma} \cdot \left\langle Linear(\hat{F}^{1/8}_{vis}), Linear(\hat{F}^{1/8}_{ir}) \right\rangle$, where $Linear(\cdot)$ is the linear layer, and $\gamma$ is the temperature parameter. The matching probability matrix is obtained by: $P_{k \in (0,1)}(i,j) = softmax(S(i,\cdot))_j$. Using the threshold $\theta_c$, high-confidence elements are filtered out to obtain coarse-level matches $M_c$.

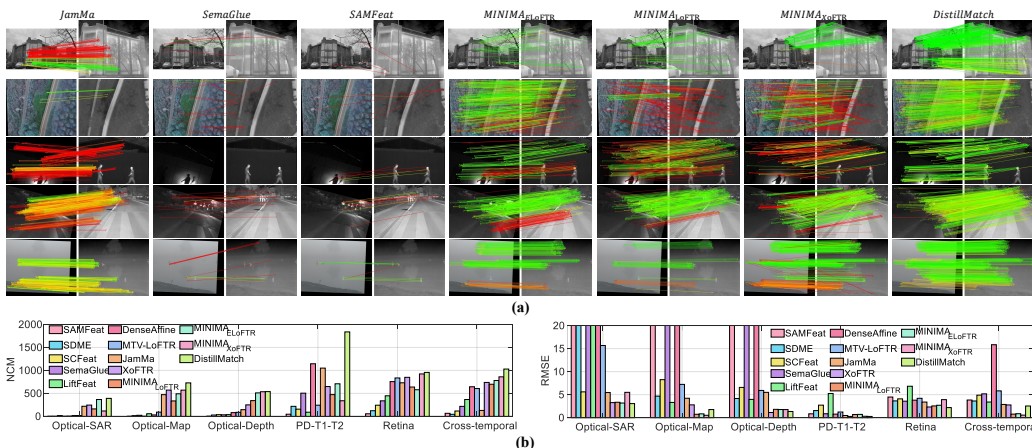

**(a)**

**(b)**

Figure 3: The qualitative and quantitative results of image matching. (a) Comparison experimental results for JamMa, SemaGlue, SAMFeat, MINIMA$_{ELoFTR}$, MINIMA$_{LoFTR}$, MINIMA$_{XoFTR}$ and DistillMatch (left to right), using images from the UAV remote sensing images, indoor scenes, nighttime conditions, haze and mist scenes (top to bottom). (b) The quantitative comparison results for zero-shot experiments of unknown modalities.

**Fine-level Matching Module (FMM):** FMM refines matches based on $M_c$ and the $1/2$ and $1/4$ scale features. As shown in Figure 2 (c), it first preprocesses $F^{1/2}$ and $F^{1/4}$ to improve feature interaction. Then, it extracts local windows of $1 \times 1$, $3 \times 3$, and $5 \times 5$ from the preprocessed features and performs a series of concatenation, self-attention, cross-attention, and splitting operations to pass information among these windows. For each $(\tilde{i}, \tilde{j})$ in $M_c$, it computes the similarity matrix $S^f$ between the processed windows $\{\hat{f}_{vis}^{5 \times 5}, \hat{f}_{ir}^{5 \times 5}\}$ and applies double softmax to obtain the fine-level match probability matrix $P^f$: $P^f(i,j) = soft\max(S^f(i, \cdot))_j \cdot soft\max(S^f(\cdot, j))_i$. Matches with $P^f(i,j) > \theta_f$ are selected as the fine-level matches $M_f$.

**Subpixel Refinement Module (SRM):** SRM refines fine-level matches to subpixel accuracy. As Figure 2 (d) shows, it concatenates $\{\hat{f}_{vis}^{5 \times 5}, \hat{f}_{ir}^{5 \times 5}\}$ at fine-level match $(\hat{i}, \hat{j})$ and predict local subpixel offsets by: $\{\delta_{vis}, \delta_{ir}\} = \mathrm{Tanh}(MLP(\hat{f}_{vis}^{5 \times 5} | \hat{f}_{ir}^{5 \times 5}))$ for each match. Adding these offsets to the coordinates of $(\hat{i}, \hat{j})$ to obtain subpixel-level matches: $\{\hat{C}_{vis}, \hat{C}_{ir}\} = \{C_{vis} + \delta_{vis}, C_{ir} + \delta_{ir}\}$, where $\{C_{vis}, C_{ir}\}$ is the coordinate of $(\hat{i}, \hat{j})$ before SRM.

### 3.5 IMAGE TRANSLATION FOR DATA AUGMENTATION

Current research suffers from the lack of large-scale visible-infrared image datasets from same scenes, and the high cost of manual annotation of matching landmarks. These factors constrain restrict improvements in multimodal matching tasks. To address this, we propose a visible-to-infrared image translation framework (V2I-GAN). V2I-GAN directly leverages mature benchmark datasets from visible image matching domains to synthesize abundant paired <visible, pseudo-infrared> data with correspondence annotations. Critically, as image translation preserves geometric structures without deformation or viewpoint changes, the synthesized data faithfully inherits both matching labels and scene diversity from the original datasets.

Based on PearlGAN's framework Luo et al. (2022); Zhu et al. (2017), we construct V2I-GAN for visible-to-infrared image translation, and train it on FMB dataset Liu et al. (2023). The architecture has two generators ($G_{VI}$, $G_{IV}$) and two discriminators ($D_V$ and $D_I$). Specifically, $G_{VI}$ transforms $I_{vis}$ into $I_{ir}^{pse}$, while $G_{IV}$ does the opposite. $D_I$ distinguishes real $I_{ir}$ from pseudo $I_{ir}^{pse}$ (from $G_{VI}$), whereas $D_V$ distinguishes real $I_{vis}$ from pseudo $I_{vis}^{pse}$ (from $G_{IV}$). The generator uses an encoder-decoder structure. The encoder extracts multi-scale texture information by convolutional layers and down-sampling blocks, while the decoder reconstructs target-domain images via up-sampling modules and feature fusion blocks. Critically, we integrate STFA in the encoder to aggregate features from DINOv2, significantly enhancing semantic comprehension of original image and the semantic consistency of generated images. Furthermore, we add a structured gradient alignment loss between input image and its semantic segmentation map to further enhance the semantic consistency. Fig-

Figure 4: The qualitative results of image translation and zero-shot experiments. (a) The qualitative results of image translation. Column 1 and 5 are visible images. Column 2 and 6 are infrared images. Column 3 and 7 are the translation results of PearlGAN. Column 4 and 8 are the translation results of V2I-GAN. (b) The qualitative results for zero-shot experiments, using images of optical-depth, optical-map, optical-optical, optical-SAR, PD-T1-T2 and retina (left to right). Line 1 are the results of MINIMA$_{\text{XoFTR}}$. Line 2 are the results of DistillMatch.

ure 4 (a) shows the image translation results. V2I-GAN's pseudo-infrared images are more like real infrared images than PearlGAN's.

### 3.6 SUPERVISION

The loss function of DitillMatch consists of the knowledge distillation loss in Equation (4), cross-entropy loss in Equation (5) and matching loss. The matching loss mainly consists of three parts:

**Coarse-level Matching Loss:** we use focus loss (FL) to supervise the matching probability matrix $P_{k \in (0,1)}$ in CMM:

$$L_c = \alpha \cdot FL(P_0, \hat{P}_0) + \beta \cdot FL(P_1, \hat{P}_1), \quad (6)$$

where $\hat{P}_0$ and $\hat{P}_1$ are the GT matching matrices for CMM. $\alpha$ and $\beta$ are the weights for balancing.

**Fine-level Matching Loss:** We design the fine-level matching loss to supervise $P^f$ in FMM:

$$L_f = \frac{1}{M_c} \sum_{(\hat{i},\hat{j}) \in M_c} FL(P^f_{\hat{i},\hat{j}}, \hat{P}^f_{\hat{i},\hat{j}}), \quad (7)$$

where $\hat{P}^f_{\hat{i},\hat{j}}$ is the GT fine-level matching matrix for $(\hat{i}, \hat{j})$.

**Subpixel Refinement Loss:** Given predicted matches' homogeneous coordinates $(\hat{x}_{vi}, \hat{x}_{ir})$, the subpixel refinement loss is computed by symmetric polar distance function:

$$L_{sub} = \frac{1}{|M_c|} \sum_{(\hat{x}_{vi}, \hat{x}_{ir})} \left\| \hat{x}_{vi}^T E \hat{x}_{ir} \right\|^2 \left( \frac{1}{\|E^T \hat{x}_{vi}\|_{0:2}^2} + \frac{1}{\|E \hat{x}_{ir}\|_{0:2}^2} \right), \quad (8)$$

where $E$ is the GT essential matrix from the camera pose. $\|v\|_{0:2}$ denotes the first two elements of the vector $v$. The total matching loss is: $L_{match} = \lambda_c L_c + \lambda_f L_f + \lambda_{sub} L_{sub}$. The overall loss is: $L_{total} = \lambda_{KD} L_{KD} + \lambda_{ce} L_{ce} + L_{match}$.

## 4 EXPERIMENTS

### 4.1 IMPLEMENTATION DETAILS

In training, we employ the MegaDepth dataset Li & Snavely (2018) as our benchmark. For data augmentation, we perform randomized adjustments to hue, saturation, and value intensities across input images, and leverage V2I-GAN to translate one image from each pair into pseudo-infrared image. Training is conducted using the AdamW optimizer with a learning rate of $6 \times 10^{-3}$, a batch size of 1, a total of 20 epochs, and 120 hours of training on 3 NVIDIA GeForce RTX 4090 GPUs. The thresholds in the matching network are set to: $\theta_c = 0.3$, $\theta_f = 0.1$. The settings in the loss function are set to: $\lambda_c = 0.5, \lambda_f = 0.3, \lambda_{sub} = 10^4, \lambda_s = 1, \lambda_{ac}^{vis} = \lambda_{ac}^{ir} = 0.25, \alpha = 100, \beta = 0.5, \gamma = 0.25, \lambda_{KD} = 0.1, \lambda_{ce} = 0.1$.

### 4.2 RELATIVE POSE ESTIMATION

**Dataset and Evaluation Metrics:** To evaluate the performance of DistillMatch for relative pose estimation in visible-infrared images, we test it on the METU-VisTIR dataset Tuzcuoğlu et al. (2024). DistillMatch processes the input images and generates matched point pairs. We use RANSAC Fischler & Bolles (1981) with a threshold of 3 to filter correct matching point pairs. During testing, the longer image side is set to 640 pixels to standardize sizes.

Table 1: Quantitative results of relative pose estimation in METU-VisTIR dataset, and the values to the left and right of '/' are the results for cloud-cloud and cloud-sunny scenarios respectively (bold fonts indicate the maximum values).

| Method | AUC of cloud-cloud and cloud-sunny | | |
|---|---|---|---|
| | @5° | @10° | @20° |
| SAMFeat | 0.084/0.083 | 0.141/0.312 | 0.323/0.931 |
| SDME | 0/0 | 0/0 | 0.198/0.222 |
| SCFeat | 0.069/0.335 | 0.489/1.404 | 2.400/4.284 |
| SemaGlue | 0.035/0.092 | 0.248/0.416 | 1.105/1.559 |
| LiftFeat | 0.131/0.173 | 0.561/0.699 | 2.176/2.818 |
| DenseAffine | 1.465/2.094 | 3.656/5.900 | 8.057/12.21 |
| MTV-LoFTR | 0.086/0.220 | 0.391/0.6128 | 1.800/2.307 |
| JamMa | 0.058/0.029 | 0.571/0.389 | 2.877/2.334 |
| XoFTR | 18.39/9.523 | 33.18/22.09 | 48.43/36.83 |
| MINIMA$_{LoFTR}$ | 19.15/10.47 | 35.78/24.84 | 52.29/41.98 |
| MINIMA$_{ELoFTR}$ | 6.872/5.248 | 17.79/14.11 | 35.10/28.59 |
| MINIMA$_{XoFTR}$ | 22.47/10.68 | 38.95/25.50 | 55.30/43.33 |
| DistillMatch | **23.13/12.45** | **41.10/26.86** | **58.41/44.47** |

We evaluate the methods independently on cloudy-cloudy and cloudy-sunny scenarios of the dataset. We use the area under curve (AUC) at 5°, 10° and 20° thresholds as evaluation metrics, measuring the maximum angular deviation from the GT in rotation and translation. We compared DistillMatch with the following publicly available methods: SAMFeat Wu et al. (2023), SDME Zhang & Ma (2024), SCFeat Cadar et al. (2024), SemaGlue Zhang et al. (2025b), LiftFeat Liu et al. (2025), DenseAffine Sun et al. (2025), MTV-LoFTR Liu et al. (2022), JamMa Lu & Du (2025), XoFTR Tuzcuoğlu et al. (2024), MINIMA$_{LoFTR}$, MINIMA$_{ELoFTR}$ Wang et al. (2024) and MINIMA$_{XoFTR}$ Ren et al. (2025).

**Results:** As shown in Table 1, DistillMatch achieves significantly higher AUC than other algorithms at all thresholds for the cloudy-cloudy and cloudy-sunny datasets. The performance on the cloudy-sunny dataset is lower than on the cloudy-cloudy dataset, due to increased image feature variation from light and temperature differences, which makes matching and pose estimation more challenging. Figure 3 (a) illustrates the qualitative results.

### 4.3 HOMOGRAPHY TRANSFORMATION ESTIMATION

**Dataset and Evaluation Metrics:** To evaluate the homography estimation performance of DistillMatch, we conducted experiments on four visible-infrared datasets covering distinct scenarios: (1) UAV remote sensing images Liu et al. (2022), (2) indoor scenes SMT/COPPE/Poli/UFRJ (2021), (3) nighttime conditions González et al. (2016), and (4) haze and mist scenes Xie & Jin (2023). We randomly generate a unique homography matrix and apply it as GT to the original image. The homography matrices include random translations of $[-10\%, 10\%]$, rotations of $[-20, -20]$, scaling of $[0.8, 1.2]$, shear angles of $[-0.1, 0.1]$, and perspective transformations of $[-0.003, 0.003]$. For UAV remote sensing dataset, we evaluate matching performance by calculating the mean reprojection error of four corner points, adopting AUC under thresholds of 3, 5 and 10 pixels. For the other datasets, AUC is computed at thresholds of 5, 10 and 20 pixels.

**Results:** As evidenced by Table 2 and Figure 3 (a), DistillMatch achieves significantly higher AUC values than competing methods across most thresholds on all datasets, with the performance gap widening progressively as thresholds increase. Figure 3(a) demonstrates that DistillMatch precisely aligns feature points between source and target images. This alignment preserves geometric consistency and structural integrity in transformed images despite scale variations, viewpoint distortions, and rotational changes.

### 4.4 ZERO-SHOT EXPERIMENTS OF UNKNOWN MODALITIES

**Dataset and Evaluation Metrics:** In addition to matching visible and infrared images, we also conducted zero-shot matching on several unknown modalities, including: (1) optical-SAR image pairs, (2) optical-map image pairs, (3) optical-depth image pairs Li et al. (2023), (4) pairwise combinations of PD, T1, and T2 images, (5) retina image pairs, and (6) cross-temporal image pairs Jiang et al. (2021). The evaluation metrics are: (1) Number of Correct Matches (NCM): A match is accepted as correct if its residual under the GT transformation is less than 5 pixels. (2) Root Mean Square Error (RMSE): The RMSE between the matches extracted by the algorithm and those under the GT transformation.

Table 2: Quantitative results of homography estimation in visible-infrared dataset. The best and second of each category are masked as bold and underline, respectively.

| Method | UAV | | | Indoor | | | Night | | | Haze | | |
|---|---|---|---|---|---|---|---|---|---|---|---|---|
| | @3px | @5px | @10px | @5px | @10px | @20px | @5px | @10px | @20px | @5px | @10px | @20px |
| SAMFeat | 3.666 | 10.85 | 24.43 | 0.384 | 0.666 | 1.122 | 0 | 0 | 0.561 | 0 | 1.462 | 3.710 |
| SDME | 4.327 | 10.85 | 21.13 | 0.309 | 0.639 | 1.090 | 0 | 0.279 | 1.272 | 0.657 | 0.945 | 2.283 |
| SCFeat | 5.799 | 16.68 | 36.43 | 0.328 | 2.406 | 12.97 | 0 | 0.406 | 6.533 | 0 | 0.922 | 4.929 |
| SemaGlue | 0.567 | 1.416 | 4.365 | 0.384 | 0.886 | 2.003 | 0.397 | 0.476 | 1.015 | 1.004 | 3.377 | 7.043 |
| LiftFeat | 8.732 | 24.28 | 45.34 | 0.870 | 2.978 | 11.69 | 0 | 1.644 | 10.49 | 0 | 1.047 | 6.477 |
| DenseAffine | 5.626 | 11.95 | 21.25 | 0 | 0 | 0 | 0 | 0.281 | 1.213 | 0 | 1.263 | 3.142 |
| MTV-LoFTR | 16.75 | 29.20 | 44.91 | 0.314 | 1.707 | 5.940 | 0 | 2.248 | 11.40 | 0.687 | 2.133 | 6.249 |
| JamMa | 0 | 0.878 | 8.676 | 0.450 | 6.275 | 19.74 | 0 | 2.965 | 15.57 | 0 | 1.250 | 2.703 |
| XoFTR | 16.93 | 35.88 | 59.33 | 2.755 | 15.62 | 29.66 | 0.363 | 3.152 | 21.17 | 6.214 | 24.66 | 45.43 |
| MINIMA$_{\text{LoFTR}}$ | 17.32 | 35.01 | 58.40 | 2.867 | 16.63 | 33.50 | 0 | 3.026 | 20.84 | 3.606 | 13.83 | 30.43 |
| MINIMA$_{\text{ELoFTR}}$ | 14.27 | 32.05 | 57.44 | 2.978 | 15.23 | 32.37 | 0 | 2.641 | 17.82 | 3.516 | 17.32 | 38.47 |
| MINIMA$_{\text{XoFTR}}$ | 19.58 | 37.65 | 60.37 | 4.793 | 18.44 | 29.46 | 0.347 | 3.256 | **22.81** | 7.719 | 26.77 | **51.35** |
| DistillMatch | **20.53** | **40.12** | **64.62** | **5.257** | **22.94** | **43.33** | **0.466** | **3.585** | 22.19 | **9.208** | **28.99** | 51.07 |

**Results:** Quantitative results are shown in Figure 3 (b). Due to the large modality gap and extreme difficulty of optical-SAR image pairs, most algorithms perform poorly. Nevertheless, our Distill-Match still has advantage. On optical-map and cross-temporal image pairs, DistillMatch slightly lags behind MINIMA$_{\text{XoFTR}}$ in terms of RMSE. However, DistillMatch achieves leading NCM across all datasets, demonstrating its robust matching capability even on unknown modalities. We attribute this primarily to the generalizable representation power of DINOv2-distilled features, and the cross-modal correlation enhancement by the CEFG. This indicates that DistillMatch possesses strong extensibility, and only needs to adapt the image translation algorithm's modality to handle diverse multimodal matching tasks. Qualitative results in Figure 4 (b) further validate that DistillMatch can establish a high quantity and proportion of correct matches on real-world multimodal image pairs.

## 4.5 ABLATION STUDY

To verify the effectiveness of Distill-Match's modules and data augmentation, we perform the ablation experiments in METU-VisTIR dataset with results in Table 3. SAA and CAA are the spatial and channel attention aggregation module. KD-VFM is feature extraction module based on knowledge distillation of VFM. V2I-GAN indicates data augmentation with V2I-GAN. A checkmark shows a module's presence. The first line is the result of baseline. The second and third lines directly aggregate the VFM features

Table 3: Ablation study of DistillMatch. All experiments are performed in the cloud-sunny scenarios of the METU-VisTIR dataset.

| SAA | CAA | KD-VFM | CEFG | V2I-GAN | AUC |
|---|---|---|---|---|---|
| | | | | | 20.05/35.94/51.98 |
| ✓ | | | | | 21.11/37.30/52.40 |
| ✓ | ✓ | | | | 21.71/38.58/54.38 |
| ✓ | ✓ | ✓ | | | 22.26/40.55/56.94 |
| ✓ | ✓ | ✓ | ✓ | | 23.12/40.24/57.40 |
| ✓ | ✓ | ✓ | ✓ | ✓ | 23.13/41.10/58.41 |

without knowledge distillation. The results show that the absence of either component degrades matching performance, underscoring their importance for cross-modal feature learning.

## 5 CONCLUSION

In this study, we propose a multimodal image matching method named DistillMatch. By leveraging knowledge distillation from VFM, it tackles modal differences and data scarcity. DistillMatch uses a lightweight student model to extract high-level semantic features from VFM for multimodal matching, and introduces a CEFG to retain modality-specific information and boost the model's understanding of cross-modality correlations. Moreover, to enhance the model's generalization ability, we design V2I-GAN for visible-to-infrared image translation as data augmentation. Experiments demonstrate that DistillMatch outperforms state-of-the-art algorithms on public datasets.

## 6 REPRODUCIBILITY STATEMENT

To support the reproducibility of our work, we have taken the following measures:

(1) Details of our proposed algorithm are fully described in Section 3 of the main paper, with implementation details provided in Section 4.1.

(2) Anonymized source code, including training scripts and evaluation procedures, is available are available at `https://anonymous.4open.science/r/DistillMatch-503A` and `https://anonymous.4open.science/r/V2I-GAN-E3B3`.

(3) The meanings of the mathematical symbols used in paper are shown in Table 4 of Appendix A.1.

(4) All datasets used in our experiments are publicly available; The detailed pre-processing procedure has been elaborately described in the Appendix A.2.

(5) The configuration and details of the training can be found in Appendix A.3.

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

# A  APPENDIX

## A.1  LIST OF MATHEMATICAL SYMBOLS

The meanings of the mathematical symbols used in the article are shown in Table 4.

| | |
|---|---|
| $F_{Res}^{1/2}, F_{Res}^{1/4} and F_{Res}^{1/8}$ | Features processed by ResNet at $1/2$, $1/4$ and $1/8$ resolution. |
| $F_{DINO}$ | Features processed by DINOv2 |
| $F_{stu}$ | The refined features output by the student model |
| $F_{tea}$ | The features output by the teacher model, and $F_{tea} = F_{DINO}$ |
| $F_{vis/ir}^{s}$ | The shallow features output by Restormer |
| $\hat{f}_{vis/ir}^{clc}$ | The feature output by modality category representation heads |
| $\widetilde{F}_{vis/ir}$ | The category-enhanced features |
| $\widetilde{F}_{Res}^{1/8}$ | The enhanced texture features |
| $F_{T}^{CAA}$ | The features after alignment in the channel dimension |
| $F_{T}^{SAA}$ | The features after spatial dimension alignment |
| $F_{T}^{STFA}$ | The features output by STFA module. |
| $S$ | The similarity matrix between $\{F_{vis}^{1/8}, F_{ir}^{1/8}\}$ |
| $P_{k \in (0,1)}, P^{f}$ | The coarse-level and fine-level matching probability matrix |
| $M_c, M_f$ | The final coarse-level and fine-level matching set |
| $S^{f}$ | The similarity matrix between $\{\hat{f}_{vis}^{5 \times 5}, \hat{f}_{ir}^{5 \times 5}\}$ |
| $\theta_c, \theta_f$ | The threshold for coarse-level and fine-level matching |
| $\{\delta_{vis}, \delta_{ir}\}$ | The local subpixel offsets for each match |
| $C_*, \hat{C}_*$ | $(\hat{i}, \hat{j})$ coordinates before and after subpixel refinement |

Table 4: List of Symbols.

## A.2  DATASET SETUP

Two parts of data are needed for training and testing DistillMatch, the original dataset, i.e., MegaDepth and METU-VisTIR, and the offline generated dataset indices. The dataset indices store scenes, image pairs, and other metadata within each dataset used for training. We use depth maps provided in the original MegaDepth dataset as well as undistorted images, corresponding camera intrinsics and extrinsics preprocessed by D2-Net. During the training phase, we use V2I-GAN to randomly perform image translation on one of the images in the image pair.

## A.3  TRAINING CONFIGURATION AND DETAILS

The knowledge distillation and matching process training of DistillMatch are carried out simultaneously, without any pre-training and fine-tuning steps. This online distillation method is beneficial for learning features in DINOv2 that are more suitable for multimodal image matching. Detailed training environment details and configurations can be found at: `https://anonymous.4open.science/r/DistillMatch-503A/requirements.txt` and `https://anonymous.4open.science/r/DistillMatch-503A/environment.yaml`.

