# OpenReview forum: "DistillMatch: Leveraging Knowledge Distillation from Vision Foundation Model for Multimodal Image Matching"
_ICLR.cc/2026/Conference — ICLR 2026 Conference Withdrawn Submission_

### Official Review · Reviewer_prnt · 2025-10-28

**Soundness:** 3
**Presentation:** 2
**Contribution:** 2
**Rating:** 6
**Confidence:** 3

**Summary:**

This paper presents DistillMatch, a novel framework for multimodal image matching that leverages knowledge distillation from Vision Foundation Models (VFMs) such as DINOv2. The method aims to bridge modality discrepancies (e.g., visible vs. infrared imagery) and mitigate data scarcity by transferring high-level semantic representations from a pretrained VFM (teacher) to a lightweight student model. To further enhance cross-modal understanding, the authors propose a Category-Enhanced Feature Guidance (CEFG) module that injects modality-specific category information into features from another modality. In addition, a V2I-GAN framework is introduced for data augmentation via visible-to-infrared translation. Experimental results on multiple public datasets demonstrate that DistillMatch achieves superior performance compared to state-of-the-art methods across various tasks, including relative pose estimation, homography estimation, and zero-shot matching.

**Strengths:**

1.	The paper introduces an effective strategy to transfer semantic knowledge from a pretrained VFM to a lightweight student network, producing generalizable and robust cross-modal representations without requiring extensive labeled datasets.

2.	The proposed Category-Enhanced Feature Guidance module effectively integrates modality-specific category representations into features from another modality, enhancing the model’s ability to capture meaningful cross-modal correlations.

3.	The proposed GAN-based visible-to-infrared translation framework addresses the scarcity of annotated multimodal datasets by generating geometrically consistent pseudo-infrared pairs, improving training diversity.

4.	DistillMatch achieves consistent improvements over strong baselines in multiple benchmark tasks, including both supervised and zero-shot scenarios. The results demonstrate strong generalization and adaptability to diverse modalities and imaging conditions.

**Weaknesses:**

1.	The writing is at times difficult to follow. For example, the abstract and introduction omit descriptions of key components such as the Coarse-level Matching Module (CMM), Fine-level Matching Module (FMM), and Subpixel Refinement Module (SRM), making it challenging for readers to grasp the overall system design early on.

2.	Essential terms and modules such as VFM and STFA are not defined or introduced in the Introduction, reducing accessibility for readers not deeply familiar with these concepts.

3.	The inclusion of a KL divergence loss between feature embeddings (F_tea and F_stu) lacks clear justification. The paper should explain why this distribution-level alignment is meaningful in the context of feature distillation.

4.	The training loss combines multiple components with numerous weighting coefficients. However, the paper does not provide rationale or sensitivity analysis for these trade-off parameters. A brief discussion or ablation would help clarify their impact.

5.	The method is evaluated only with DINOv2 as the teacher. It remains unclear whether similar gains would hold when using other VFMs such as CLIP, SAM, or EVA. Evaluating multiple teachers would strengthen claims of generalizability.

6.	Since DINOv2 is trained primarily on large-scale visible images, it is uncertain how reliable the distilled features are when applied to modalities such as infrared, retina, or depth.

7.	The paper lacks ablation studies isolating the impact of losses or modules related to the CMM, FMM, and SRM. This omission makes it difficult to quantify how each contributes to overall performance.

8.	The description of V2I-GAN is terse and lacks supporting figures or algorithmic clarity. Specifically, it is unclear how the STFA module is integrated into the encoder, and the definition and intuition behind the structured gradient alignment loss should be elaborated.

9.	The paper presents qualitative examples comparing V2I-GAN with PearlGAN but does not include quantitative metrics (e.g., FID, LPIPS, or domain similarity scores) to substantiate claims of improved realism or domain transfer fidelity.

10.	The introduction repeats background material and lacks a clear logical progression from problem statement to motivation and contributions.

**Questions:**

See the weaknesses.

---

### Official Review · Reviewer_fXMe · 2025-10-29

**Soundness:** 2
**Presentation:** 2
**Contribution:** 2
**Rating:** 2
**Confidence:** 4

**Summary:**

This paper address multimodal image matching via distilling useful knowledge from pretrained vision-language models (DINO v2). To make it, a lightweight student model is learned by online knowledge distiilation. Besides, category-enhanced feature guidane module (CEFG) , and semantic and texture feature aggregation module (STFA) are developed in the proposed matching framework namely DistillMatch. Experimental results show the effectiveness of these key components.

**Strengths:**

Image matching is an important and fundamental task in computer vision. It is convincing to leverage VLMs to advance multimodal image matching.

The experiments are comprehensive with both quantitative and qualitative evaluations.

**Weaknesses:**

Although some implementation details have been provided, the writing needs more improvements. Some descriptions are unclear and hard to understand. For example, it is not easy to capture the technial contributions in this work. Besides, what are the relations between the key components such as CEFG and STFA.

It is hard to understand the complex processes in CEFG and STFA. Are there any empirical insights behind these components?

The knowledge distillation algorithm used in this work is not novel at all. In addition, the three loss weights in Eq.(4) are redundant. Perhaps, only two of them are enough for parameter tuning.

Some implementation details are missing.

I am confused by the construction of semantic and texture feature aggregation module. More detailed description and motivation are needed.

**Questions:**

Why both ResNet and ViT are both used for feature extraction. It lacks convincing explanation in the paper.

Why the images are downsampled to 7/8? Is there any important evidence?

In Line 215, what is learnable category feature? Or where is it from?

Whether the proposed method can be applicable to other VLMs as well?

---

### Official Review · Reviewer_Bn9Q · 2025-10-29

**Soundness:** 2
**Presentation:** 1
**Contribution:** 2
**Rating:** 0
**Confidence:** 3

**Summary:**

This paper addresses multi-modal image matching by proposing DistillMatch, a knowledge distillation framework that leverages vision foundation models. The authors use a teacher-student architecture where a pre-trained vision model serves as the teacher to extract high-level features, while the student model is enhanced with additional modality information for improved generalization across different imaging modalities. The paper additionally proposes to add a GAN based approach for augmentation

**Strengths:**

The paper studies an important problem in multi-modal alignment with practical applications. It introduces a variant  of knowledge distillation to enhance cross-modal image matching, enabling better alignment and understanding across different data modalities. Additionally, it tackles the challenge of data scarcity by incorporating effective data augmentation strategies, thereby improving the model’s robustness and generalization.

**Weaknesses:**

## Major Concerns
1)Clarity and Technical Details

Line 72 & Introduction: The paper introduces V2I-GAN without proper explanation. What does V2I-GAN stand for? How does it translate visible to pseudo-infrared images? This critical component needs clear definition and technical description.
Line 70: STFA is mentioned without definition. Is this "Student Teacher Feature Aggregation"? Please clarify all acronyms upon first use.
Line 160: The superscripts 1/2, 1/4, 1/8 appear to refer to scales but this should be explicitly stated.

2. Technical Methodology Issues

Distillation Process: With multiple resolutions in the distillation process, have the authors considered channel-wise normalization? This could be crucial for stable training across different scales.
Architecture Details: The transition from student model input (P=1600) to F_student (C_4=384) needs clarification. How exactly does the downsampling process work?
Loss Function: What additional information does the Gram matrix provide beyond MSE loss? The motivation for this choice needs better justification.

3. Experimental Evaluation

Comparison with Related Work: While the related work section covers relevant literature, the paper lacks clear comparison showing how the proposed method improves upon existing approaches. Quantitative comparisons with state-of-the-art methods are essential.
Ablation Study: The ablation study only examines individual components but misses the impact of different hyperparameters. This limits understanding of the method's sensitivity and robustness.

Minor Issues

Writing Quality: The paper would benefit from careful proofreading. Several sentences are unclear or grammatically incorrect.
Figure Quality: Ensure all figures are clearly labeled and referenced in the text.

**Questions:**

1)How does DistillMatch specifically improve upon existing cross-modal matching methods quantitatively?
2)What is the computational overhead of the multi-resolution distillation process? This comes with respect to GAN which introduces an additional computational overhead ?
3)Have you tested the method on other modality pairs beyond visible-infrared?
4)What happens to performance when V2I-GAN augmentation is removed entirely? What about the aspect of alternate augmentations like Autoaugment or so

---

### Official Review · Reviewer_Vuwx · 2025-11-04

**Soundness:** 3
**Presentation:** 3
**Contribution:** 3
**Rating:** 6
**Confidence:** 4

**Summary:**

This paper proposes DistillMatch, a novel method for multimodal image matching designed to overcome the inherent differences between modalities and the prevalent scarcity of high-quality annotated datasets. The core innovation lies in utilizing knowledge distillation from a Vision Foundation Model (VFM) to train a lightweight student model, coupled with specialized modules to manage modality-specific information and enhance feature aggregation. Extensive experiments demonstrate that DistillMatch significantly outperforms existing state-of-the-art algorithms across various matching tasks and exhibits strong zero-shot generalization.

**Strengths:**

* Usage of vision foundation models like Dinov2 to guide knowledge distillation process for lightweight student model through multi-component loss.
* Retaining modality specific information through Category-Enhanced feature guidance module.
* Robust feature integration through STFA module relying on both channel and spatial attention aggregation.
* Data augmentation strategy for visible to infrared image translation through V2I-GAN framework
* State of the art results across multiple benchmarks for relative pose estimation, homography transformation estimation.

**Weaknesses:**

* How is the quality of the synthetic infrared data evaluated ? Further, how does the quality of synthetic data vary with the inclusion of recent diffusion models e.g. DiffV2IR (https://arxiv.org/pdf/2503.19012) ?
*In terms of the training data, what fraction of samples were obtained through synthetic data augmentation ?
* The ablation studies associated with the different components of the knowledge distillation loss (LKD) i.e. Eq 4 are not included in the Experiments section.

**Questions:**

* In terms of KD based supervision, have the authors considered other VFM models like Radio (https://arxiv.org/abs/2312.06709) for feature extraction ?

---

### Note · Authors · 2025-11-12

I have read and agree with the venue's withdrawal policy on behalf of myself and my co-authors.